# GeoAssistant: A Geospatial Vision and Language Assistant that Plugs and Learns to Use Tools for Remote Sensing

## Abstract

Vision-language models (VLMs) hold great potential for interpreting large-scale remote sensing (RS) archives, which are critical for applications such as environmental monitoring, disaster response, and urban planning. However, general-purpose VLMs primarily focus on optical imagery, perform poorly on RS tasks, and existing RS-specific VLMs still struggle with fine-grained understanding. To address these limitations, we propose GeoAssistant, a tool-augmented multimodal assistant tailored for RS scenarios. GeoAssistant interprets user instructions, autonomously determines whether to invoke external tools, and synthesizes their outputs to generate precise responses. A key innovation of our approach is its capability to process both optical and Synthetic Aperture Radar (SAR) imagery, enabling a wide range of tasks, including visual grounding, object detection, segmentation, and multifaceted reasoning. To support this, we construct the first cross-domain, tool-augmented instruction dataset for RS, addressing the critical challenge of task-specific data scarcity. We also introduce GeoAssistBench, a comprehensive benchmark for cross-domain, multi-task dialogue in RS, and use it to evaluate GeoAssistant. Our results show that GeoAssistant consistently outperforms existing RS-specific VLMs across diverse tasks, demonstrating its practical value for real-world RS applications.

## 1 Introduction

Recent advancements in Vision-Language Models (VLMs) have demonstrated remarkable success in the natural image domain. These models enable unified visual understanding, capable of performing diverse tasks such as classification, localization, visual question answering, and dense captioning (Liu et al., 2023; 2024; Chen et al., 2024; 2023b). Their strong conversational and instruction-following capabilities have paved the way for general-purpose multimodal assistants (Devlin et al., 2019; Achiam et al., 2023a; Chen et al., 2023a; Bai et al., 2023; Chen et al., 2023c; Qwen Team, 2025). This success has motivated efforts to bring such capabilities into the Remote Sensing (RS) domain. However, creating a truly practical, general-purpose RS assistant requires overcoming two critical frontiers largely unaddressed by current models.

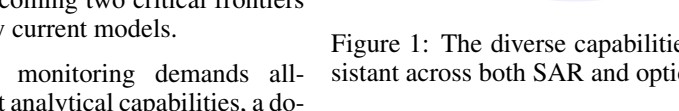

Figure 1: The diverse capabilities of GeoAssistant across both SAR and optical imagery.

First, resilient earth monitoring demands all-weather, day-and-night analytical capabilities, a domain where optical-only models fail. Although most research focuses on optical imagery, Synthetic Aperture Radar (SAR) data provides this vital capability. For example, during a hurricane, SAR can penetrate thick cloud cover to map flooded areas and assess infrastructure damage when optical sensors are rendered ineffective. A robust RS assistant must therefore master SAR data to ensure

operational continuity for time-critical applications like disaster management and maritime security. Second, real-world problems require the execution of complex analytical workflows, not just answering simple questions. A practical system must therefore be able not only to compose these workflows from a suite of specialized tools but also to be designed for extensibility, allowing for the integration of new capabilities as they emerge.

Despite recent progress, existing RS-VLMs still fall short of these goals. Efforts like SkyEyeGPT (Zhan et al., 2025) have focused on building large-scale instruction datasets, while others like GeoChat (Kuckreja et al., 2024) have improved region-level understanding. However, their research focus remains predominantly on optical data, and more critically, they lack a unified framework for the dynamic orchestration of specialized tools (Zhan et al., 2025).

To overcome these multifaceted challenges, we introduce GeoAssistant, a tool-augmented multimodal assistant for RS. As illustrated in Figure 1, GeoAssistant supports a broad spectrum of tasks on both SAR and optical (RGB) imagery, which we categorize into two types: (i) General Abilities, such as VQA and captioning, relying on the model's inherent knowledge, and (ii) specialized Tool Abilities, such as object detection and segmentation, enabled through external tools. Rather than functioning as a monolithic model, GeoAssistant is designed as a modular, end-to-end workflow framework that integrates perception, instruction parsing, task decomposition, tool scheduling, and response generation, while remaining extensible to future tools and capabilities.

To support this system, we construct the first large-scale instruction dataset for tool use in the RS domain, comprising over 554K samples with both optical and SAR data alongside detailed tool-use chains of thought. Furthermore, we introduce GeoAssistBench, a comprehensive benchmark suite for evaluating multi-task, cross-domain dialogue and tool-augmented reasoning in RS. Extensive experiments on GeoAssistBench demonstrate that GeoAssistant consistently outperforms existing RS-specific VLMs across diverse tasks.

In summary, our main contributions are as follows:

- We construct the first large-scale instruction dataset for tool augmentation in RS, containing over 554K instruction pairs with both optical and SAR data. This dataset enhances the model's cross-domain understanding and tool-application capabilities.
- We develop GeoAssistant, a unified, tool-augmented assistant that dynamically orchestrates specialized visual tools based on user instructions. It executes complex tasks on both optical and SAR imagery in an end-to-end manner, generating modular and extensible workflows.
- We introduce GeoAssistBench, a benchmark suite covering both optical and SAR domains, and use it to comprehensively evaluate GeoAssistant. Experimental results show that GeoAssistant consistently outperforms strong VLM and RS-specific baselines on tasks such as VQA, captioning, and tool-augmented tasks.

## 2 RELATED WORK

**The Evolution of Tool-Augmented Agents.** The long-standing pursuit of building artificial intelligence agents that integrate perception, reasoning, and action (Ruan et al., 2023) began with foundational paradigms such as symbolic agents for deliberative planning (Newell & Simon, 1976) and reactive agents for rapid responses (Brooks, 1991). The advent of machine learning, particularly deep reinforcement learning, catalyzed a shift towards data-driven decision-making, with seminal works like DQN, AlphaGo, and MAML demonstrating increasingly sophisticated strategies (Mnih et al., 2015; Silver et al., 2016; Finn et al., 2017).

More recently, the capabilities of LLMs and VLMs have enabled the development of general-purpose agents with sophisticated planning and instruction-following abilities (Ouyang et al., 2022). LLMs like GPT-4 exhibit powerful reasoning (Achiam et al., 2023b), while VLMs such as CLIP (Radford et al., 2021), BLIP (Li et al., 2022), and LLaVA (Liu et al., 2023) have advanced joint vision-language understanding. To enhance agents' capacity to perform complex tasks, researchers have focused on integrating external tools. Frameworks like ReAct (Yao et al., 2023) interleave reasoning with actions, Toolformer (Schick et al., 2023) empowers LLMs to autonomously use APIs, and LLaVA-Plus (Liu et al., 2024) extends this paradigm to vision-language tasks through a unified tool interaction framework driven by image-grounded instructions. Despite their progress, these

Table 1: A comprehensive comparison of capabilities for different RS vision-language models. CL: Classification, IC: Image Captioning, VQA: Visual Question Answer, OP: Object Positioning, PD: Panoptic Detection, OD: Object Detection, VG: Visual Grounding, SS: Semantic Segmentation, IS: Instance Segmentation, RS: Referring Segmentation.

| Models | Image level | | | | | | | Region level | | | | | Pixel level | | | External Skills | |
|---|---|---|---|---|---|---|---|---|---|---|---|---|---|---|---|---|---|
| | CL | IC | VQA | $CL_{SAR}$ | $IC_{SAR}$ | $VQA_{SAR}$ | $OP_{SAR}$ | PD | OD | VG | $PD_{SAR}$ | $VG_{SAR}$ | SS | IS | RS | Tool | Learning |
| GeoChat | ✓ | ✓ | ✓ | | | | | | | ✓ | | | | | | | |
| LHRS-Bot | ✓ | ✓ | ✓ | | | | | ✓ | ✓ | | | | | | | | |
| RSGPT | ✓ | ✓ | ✓ | | | | | | | | | | | | | | |
| EarthGPT | ✓ | ✓ | ✓ | | | | | | | ✓ | | | | | | | |
| RS-ChatGPT | ✓ | ✓ | ✓ | | | | | | | | | | | | | ✓ | |
| SkyEyeGPT | ✓ | ✓ | ✓ | | | | | | | ✓ | | | | | | | |
| EarthMarker | ✓ | ✓ | ✓ | | | | | | | | | | | | | | |
| Falcon | ✓ | ✓ | ✓ | | | | | | ✓ | ✓ | | | ✓ | | | | |
| RS-Agent | ✓ | ✓ | ✓ | | | | | ✓ | | | ✓ | | | ✓ | | ✓ | |
| **GeoAssistant (Ours)** | ✓ | ✓ | ✓ | ✓ | ✓ | ✓ | ✓ | ✓ | ✓ | ✓ | ✓ | ✓ | ✓ | ✓ | ✓ | ✓ | ✓ |

systems are predicated on the statistical patterns of natural images. Their direct application to RS is consequently hindered by a significant domain gap, where factors like extreme scale variation and sparse semantic content challenge their foundational architectural assumptions.

**Intelligent Agents in Remote Sensing.** To address this gap, several agent-like systems have been specifically developed for RS. SkyEyeGPT (Zhan et al., 2025) unifies diverse RS tasks under a single instruction-tuned interface, while LRSCLIP (Chen et al., 2025) and RS-MoE (Lin et al., 2025) explore specialized image-text alignment and modular expert selection, respectively. Other works like GeoChat (Kuckreja et al., 2024) have advanced region-level dialogue but are constrained by static task flows and an absence of tool integration. RS-Agent (Xu et al., 2024) builds upon this by incorporating tool modules, yet its reliance on fixed toolchains and predefined templates limits its adaptability to novel tasks and flexible tool compositions. Furthermore, these systems are largely confined to optical data, lacking robust SAR processing capabilities. A common thread among these pioneering efforts is the challenge of achieving dynamic, context-aware tool orchestration grounded in high-quality, multi-modal instruction data.

To provide a clear overview of the current landscape and position our work, we present a comprehensive comparison of capabilities in Table 1. The table summarizes the state-of-the-art in RS VLMs, including prominent systems like GeoChat, SkyEyeGPT, LHRS-Bot (Muhtar et al., 2024), RSGPT (Hu et al., 2023), EarthGPT (Zhang et al., 2024b), RS-ChatGPT (Guo et al., 2024), Earth-Marker (Zhang et al., 2024a), Falcon (Yao et al., 2025a), and RS-Agent (Xu et al., 2024). As Table 1 shows, while many models perform image-level tasks, few offer a complete suite of region and pixel-level abilities. Furthermore, support for SAR data and dedicated training for tool use are notably rare, motivating the design of our GeoAssistant.

Conversely, our proposed GeoAssistant is architected to overcome these specific limitations. It integrates an instruction-tuned LLM with a flexible tool orchestration framework that enables robust instruction parsing and dynamic task scheduling. Unlike previous systems dependent on static pipelines, GeoAssistant supports end-to-end, tool-aware execution grounded in domain-specific visual inputs, including both optical and SAR imagery.

## 3 GeoAssistant

GeoAssistant is designed to address a spectrum of multi-modal RS tasks, ranging from holistic image comprehension to fine-grained, spatially-aware interactions. Its operational framework is designed to strategically leverage both its intrinsic cross-domain understanding capabilities and its selective tool augmentation mechanism, thereby ensuring response robustness and precision. Figure 2 shows the overview of GeoAssistant's architecture. The tasks executable by GeoAssistant, characterized by their interaction modalities and processing mechanisms, can be principally categorized as follows:

**a) General Image Understanding and Dialogue:** This operational mode is dedicated to holistic, context-driven reasoning, processing input RS imagery in conjunction with user textual queries. It

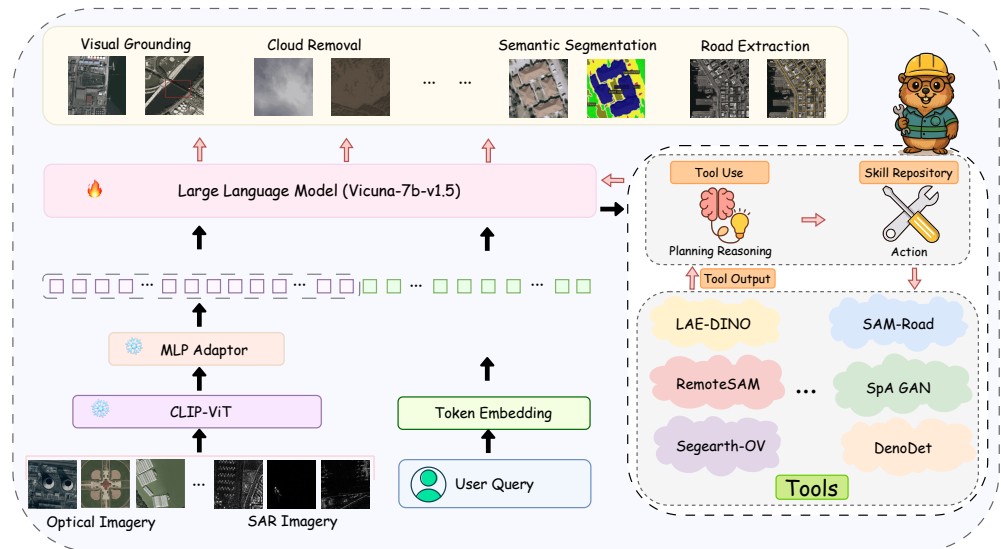

Figure 2: An overview of the GeoAssistant architecture. Our model is centered around Vicuna 7B-v1.5, which processes multi-modal inputs. The framework features a dual-pathway design: it can directly generate responses for general understanding tasks (top path), or invoke specialized external tools via a planning and reasoning module for precision-based tasks (right path).

facilitates comprehensive image-level dialogue, often dispensing with the need for pre-defined spatial coordinates. The mode's scope encompasses tasks from two primary categories: it addresses General RS Understanding by handling VQA, scene classification, and image description. Crucially, it also covers the SAR RS Understanding category, addressing SAR-specific VQA and image description.

**b) Tool-Augmented Spatial Reasoning:** This mode is engaged for tasks demanding fine-grained spatial reasoning or precise localization, which correspond to our RS Specific Tasks category and include referring object grounding, interactive segmentation, and semantic segmentation. The cornerstone of this mode is GeoAssistant's selective tool invocation mechanism. When a tool is deemed necessary, GeoAssistant seamlessly integrates it into its workflow of instruction parsing, tool selection, execution, and result synthesis to produce accurate, visually grounded outputs. This adaptive methodology ensures both flexibility in handling diverse instructions and high-fidelity results for precision-critical applications.

### 3.1 ARCHITECTURE

GeoAssistant's architecture is built upon the LLaVA-Plus framework (Liu et al., 2024), which endows the system with advanced multi-modal reasoning and dynamic tool integration capabilities for RS. This architecture integrates three core components:

(i) A Visual Backbone, typically a pre-trained Vision Transformer. For this work, we select CLIP-ViT(L-14) as the visual backbone, which has an input resolution of 336×336.

(ii) A Cross-Modal Adapter, which is a lightweight two-layer Multi-Layer Perceptron (MLP) consistent with the LLaVA architecture (Liu et al., 2023). We initialize this component using the MLP projector from LLaVA, pre-trained on CC3M-595K. This powerful initial alignment provides a robust foundation for the model's subsequent fine-tuning on domain-specific RS tasks, enabling efficient and effective vision-language fusion.

(iii) An LLM, serving as the core cognitive hub of GeoAssistant. Its primary responsibilities include interpreting user instructions, fusing multimodal inputs, performing reasoning, and generating textual responses. Crucially, it is also tasked with orchestrating the selective invocation of external

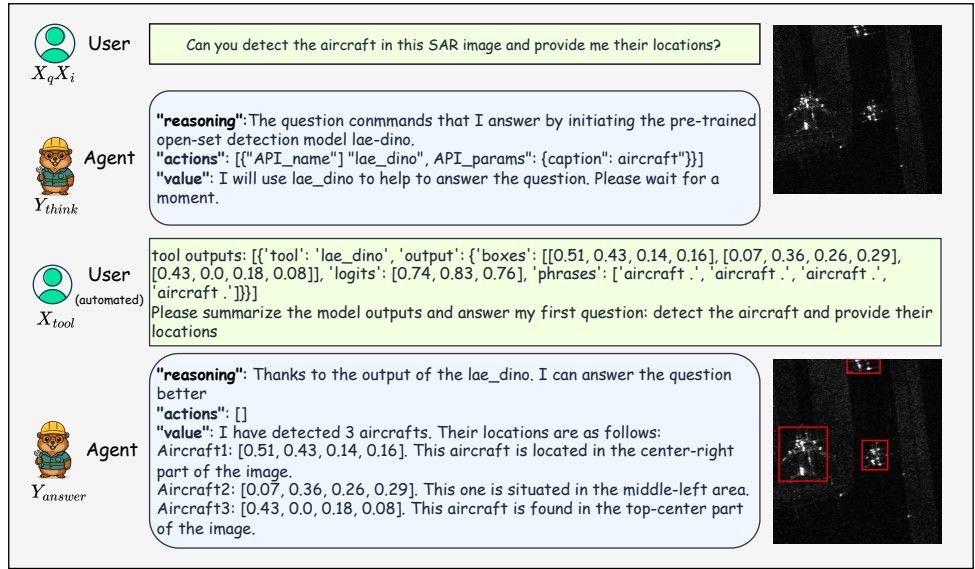

Figure 3: GeoAssistant workflow on a tool-augmented object detection task. The process unfolds in two turns. In the first turn, the agent receives the user's multimodal query ($X_q$, $X_i$) and generates a plan ($Y_{think}$) that includes its reasoning and a call to an external tool. In the second turn, the agent receives the tool's output ($X_{tool}$) and synthesizes this new information to produce the final, grounded answer ($Y_{answer}$). "User (automated)" step denotes an internal system process that simulates a second conversational turn by reformatting the tool's output and feeding it back to the agent.

specialized tools when precision is paramount. This architecture employs the open-source Vicuna-v1.5 (7B) (Zheng et al., 2023a) as this central unit.

## 3.2 WORKFLOW AND UPDATING FUNCTION

GeoAssistant operates through a structured, multi-turn dialogue workflow that unifies its direct reasoning and tool-augmented capabilities. This interaction, formalized as a two-turn process (Figure 3), is represented by the following sequence format:

$$\text{User} : X_i, \ X_q \texttt{<STOP>} \ \text{Agent} : Y_{think} \texttt{<STOP>} \tag{1}$$

$$\text{User} : X_{tool} \texttt{<STOP>} \ \text{Agent} : Y_{ans} \texttt{<STOP>} \tag{2}$$

In the first turn, the user initiates the dialogue with a multi-modal query composed of an image $X_i$ and a textual instruction $X_q$. The Agent processes this input and formulates a response $Y_{think}$, which encapsulates its internal reasoning and a planned tool action. Following this, the system executes the specified tool and returns its output as an observation, $X_{tool}$. In the second turn, this observation is provided as input to the Agent, which then synthesizes all available information to generate the final textual answer, $Y_{ans}$.

To accommodate both operational modes, the Agent's outputs ($Y_{think}$ and $Y_{ans}$) adhere to a unified format comprising three fields: *reasoning*, *actions*, and *value*. For tasks not requiring external tools, the *actions* field is null. The entire model is trained end-to-end with a standard auto-regressive objective. The training loss is exclusively computed on the Agent-generated tokens ($Y_{think}$ and $Y_{ans}$). This targeted updating strategy compels the model to learn not only *what* to respond, but also *how* and *when* to invoke tools, thereby mastering the complete reasoning and harmonized workflow from the instruction data.

## 4 RS TOOL-AUGMENTED INSTRUCTION DATASET

To serve as the foundation for training GeoAssistant, we constructed a large-scale, tool-augmented instruction dataset comprising over 554,913 samples. The detailed breakdown of our dataset is summarized in Table 2.

Table 2: Overview of the abilities of GeoAssistant and data statistics of our constructed RS tool-augmented instruction Dataset. **For optical imagery**: DOTA (Xia et al., 2018), DIOR (Li et al., 2020), FAIR 1M (Sun et al., 2022) are datasets for Panoptic Detection and Object Detection; Ris-Bench (Dong et al., 2024), OpenEarthMap is dataset (Xia et al., 2023) for Semantic Segmentation; RisBench, iSAID (Zamir et al., 2019) are datasets for Instance Segmentation and Referring Segmentation; SpaceNet (Van Etten et al., 2018), CityScale (Hetang et al., 2024) are datasets for Road Extraction; RICE is dataset for Cloud Removal (Lin et al., 2019). **For SAR imagery**: SARLANG-1M (Wei et al., 2025) is dataset for General Understanding; SARDet-100K (Li et al., 2024b) is dataset for Panoptic Detection and Visual Grounding.

|  | Abilities | Tools | Source | Size |
|---|---|---|---|---|
| **Optical Imagery Tasks** | General Understanding | - | GeoChat Instruct | 198,326 |
|  | Panoptic Detection | LAE-DINO Pan et al. (2024) | DOTA, DIOR, FAIR1M | 18,982 |
|  | Object Detection | LAE-DINO | DOTA, DIOR, FAIR1M | 12,557 |
|  | Visual Grounding | RemoteSAM Yao et al. (2025b) | RiSBench, DIOR | 16,086 |
|  | Semantic Segmentation | Segearth-OV Li et al. (2024a) | OpenEarthMap | 6,000 |
|  | Instance Segmentation | RemoteSAM | RisBench, iSAID | 13,000 |
|  | Referring Segmentation | RemoteSAM | RisBench, iSAID | 14,999 |
|  | Road Extraction | SAM-Road Hetang et al. (2024) | SpaceNet, CityScale | 1,236 |
|  | Cloud Removal | SpA GAN Pan (2020) | RICE | 2,960 |
| **SAR Tasks** | General Understanding | - | SARLANG-1M | 249,488 |
|  | Panoptic Detection | DenoDet Dai et al. (2024) | SARDet-100K | 11,710 |
|  | Visual Grounding | fintuned LAE-DINO | SARDet-100K | 9,569 |
| **Total** | - | - | - | **554,913** |

Table 3: Overview of GeoAssistBench, including datasets, size, and input/output formats.

| Task | Dataset | Size | Input/Output |
|---|---|---|---|
| Task Planning | Custom | 50 | Instruction $\rightarrow$ Tool Calls |
| VQA | RSVQA-LRBEN | 10,004 | Image + Question $\rightarrow$ Answer |
|  | RSVQA-HRBEN | 62,554 | Image + Question $\rightarrow$ Answer |
| Referring Object Detection | GeoChat-Instruct | 7,593 | Image + Referring Expression $\rightarrow$ Bounding Box |
| SAR VQA | SARDet-100K | 11,955 | Image + Question $\rightarrow$ Answer |
| SAR Image Captioning | SARLANG-1M | 6,000 | Image $\rightarrow$ Text |

### 4.1 DATA FOR GENERAL UNDERSTANDING

To cultivate the GeoAssistant's internal reasoning for tool-free scenarios, we curated a corpus for general visual understanding across both optical and SAR domains. For optical data, we refined the GeoChat Instruct dataset (Kuckreja et al., 2024), filtering out instances requiring grounding and using an LLM to generate an explicit reasoning step for each instruction-response pair. For the SAR domain, we adapted the SARLANG-1M dataset (Wei et al., 2025) by converting its image-caption pairs into a similar instruction-following format, again generating relevant questions and a corresponding thought process. This unified data structure teaches the agent to confidently handle queries based on its inherent knowledge across different modalities.

### 4.2 DATA FOR TOOL-AUGMENTED TASKS

To teach GeoAssistant how and when to invoke external tools, we developed a unified data generation pipeline and applied it to both optical and SAR source datasets. This pipeline systematically converts ground-truth labels into a two-turn dialogue format. For each source label, we prompt the Gemini 2.5 Flash (Doshi, 2025) to: (i) generate a plausible human-like query ($X_q$). (ii) produce the thought process and a structured tool call ($Y_{think}$). (iii) format the ground-truth as the tool's output ($X_{tool}$). (iv) generate a final answer summarizing the findings ($Y_{ans}$). This scalable methodology allowed us to create a comprehensive dataset for training tool-use across a diverse set of precision-critical tasks, including detection, segmentation, and grounding for both optical and SAR imagery.

## 5 EXPERIMENT

### 5.1 BENCHMARK SETUP: GEOASSISTBENCH

To systematically evaluate GeoAssistant, we introduce GeoAssistBench, a benchmark suite spanning both optical and SAR modalities across a wide range of RS tasks. GeoAssistBench is constructed from established datasets and reformulated into a unified instruction–response format with explicit tool-use chains of thought. The suite covers both general understanding and tool-augmented reasoning. The detailed information can be seen from Table 3. GeoAssistBench includes the following four types of tasks:

- **Task Planning Accuracy:** A test set of 50 QA pairs covering object detection and instance segmentation, measuring whether the agent correctly invokes the required tool sequence.
- **Visual Question Answering (VQA):** RSVQA-LRBEN and RSVQA-HRBEN (Lobry et al., 2020), covering diverse geographic regions and question types.
- **Referring Object Detection:** GeoChat-Instruct (Kuckreja et al., 2024), using Acc@0.5.
- **SAR Understanding:** ARDet-100K (Li et al., 2024b) for SAR VQA (object identification, counting, classification, positioning), and SARLANG-1M (Wei et al., 2025) for captioning. Metrics: accuracy for VQA; BLEU (Papineni et al., 2002), ROUGE-L (Lin, 2004), and CIDEr (Vedantam et al., 2015) for captioning.

### 5.2 IMPLEMENTATION, TRAINING AND SERVING

We initialize GeoAssistant using pre-trained CLIP-ViT-L/14 and LLaVA-v1.5 (7B) weights. The model undergoes full-parameter fine-tuning for 2 epochs using the AdamW optimizer, a learning rate of 2e-5, a cosine learning rate scheduler, and a global batch size of 144. All images are processed at a $336 \times 336$ resolution. The entire training process took approximately 4 days on a system equipped with 8 NVIDIA 6000 Ada GPUs. The fine-tuning curriculum is a two-stage process: an initial stage on our optical dataset, followed by a second stage on the SAR dataset. To mitigate catastrophic forgetting, the second stage's data is a mixture of SAR samples and 20% of the optical image-text pairs. During training, each tool-augmented task is guided by a unique chain-of-thought template. For deployment, the fully trained 7B GeoAssistant and its tools are served via the FastChat system (Zheng et al., 2023b) on a single NVIDIA A100 80GB GPU.

### 5.3 QUANTITATIVE RESULTS

**Task Planning Accuracy:** As shown in Table 4, GeoAssistant achieves a perfect 100% accuracy on task planning, surpassing RS-ChatGPT and RS-Agent. This highlights that our advantage stems from architectural design for tool orchestration rather than solely the underlying LLM.

Table 4: Comparison of our GeoAssitant with specialized VLMs on Task Planning Accuracy. OD: Object Detection, IS: Instance Segmentation.

| Model | OD | IS | Avg |
|---|---|---|---|
| RS-ChatGPT | | | |
| *gpt-3.5-turbo-1106* | 77.42% | 55.39% | 66.40% |
| *gpt-3.5-turbo* | 51.21% | 76.43% | 63.82% |
| *gpt-4o-mini* | 44.15% | 84.45% | 64.30% |
| RS-Agent | | | |
| *gpt-3.5-turbo-1106* | 82.78% | 79.45% | 81.11% |
| *gpt-3.5-turbo* | 90.21% | 68.89% | 79.55% |
| *gpt-4o-mini* | 94.79% | 100% | 97.39% |
| **GeoAssistant (Ours)** | **100%** | **100%** | **100%** |

**Visual Question Answering:** On both RSVQA-LRBEN and the more challenging RSVQA-HRBEN benchmarks, GeoAssistant outperforms all baselines across every metric (Table 5, Table 6). These results demonstrate robust visual reasoning and strong generalization ability.

Table 5: Comparison of our GeoAssistant with VLMs on the RSVQA-LRBEN dataset.

| Model | Presence | Comparison | Rural/Urban | Average |
|---|---|---|---|---|
| LLaVA-1.5 (Liu et al., 2023) | 55.46 | 68.20 | 59.00 | 62.77 |
| MiniGPTv2 (Chen et al., 2023a) | 55.16 | 55.22 | 39.00 | 54.96 |
| RSGPT (Hu et al., 2023) | 91.03 | 91.70 | 94.00 | 92.29 |
| GeoChat(Kuckreja et al., 2024) | 91.09 | 90.33 | 94.00 | 90.70 |
| LHRS-Bot (Muhtar et al., 2024) | 88.51 | 90.00 | 89.07 | 89.19 |
| **GeoAssistant** | **92.06** | **91.88** | **95.12** | **93.23** |

Table 6: Comparison of our GeoAssistant with VLMs on the RSVQA-HRBEN dataset. Results are reported using the Accuracy.

| Model | Presence | Comparison | Average |
|---|---|---|---|
| MiniGPTv2 (Chen et al., 2023a) | 40.79 | 50.91 | 46.46 |
| Qwen-VL (Qwen Team, 2025) | 66.44 | 60.41 | 63.06 |
| InternVL2-8B (Chen et al., 2023c) | **67.35** | 76.91 | 72.70 |
| GeoChat(Kuckreja et al., 2024) | 58.45 | 83.19 | 72.30 |
| EarthGPT (Zhang et al., 2024b) | 62.77 | 79.53 | 72.06 |
| **GeoAssistant** | 63.45 | **83.91** | **73.68** |

**Referring Object Detection:** On the GeoChat-Instruct dataset (Table 7), GeoAssistant consistently surpasses advanced VLM baselines. The gains are especially pronounced in the *Multiple* objects category, where it outperforms the strongest baseline by a large margin.

Table 7: Comparison of our GeoAssistant with VLMs on the Geochat-Instruct dataset. Results are reported using the Acc@0.5 metric.

| Model | Small | Medium | Large | Single | Multiple |
|---|---|---|---|---|---|
| MiniGPTv2 (Chen et al., 2023a) | 1.7 | 9.9 | 21.9 | 9.1 | 3.6 |
| GeoChat(Kuckreja et al., 2024) | 2.90 | 13.60 | 21.70 | 16.00 | 4.30 |
| InternVL2-8B (Chen et al., 2023c) | 7.20 | 23.76 | 31.99 | 25.77 | 9.30 |
| **GeoAssistant (Ours)** | **10.63** | **25.11** | **32.45** | **26.32** | **16.84** |

**SAR Visual Question Answering and Captioning:** GeoAssistant achieves state-of-the-art performance on SARDet-100K for VQA and SARLANG-1M for captioning. As shown in Table 8 and Table 9, it substantially outperforms baselines across all SAR VQA subtasks and delivers CIDEr scores an order of magnitude higher in SAR captioning. These results confirm its superior ability to interpret complex SAR imagery.

Table 8: Comparison of our GeoAssistant with VLMs on the SAR Captioning task. The version used here for QWEN2.5-VL and LLaVA1.5 are 7B size model.

| Model | BLEU_1 | BLEU_2 | BLEU_3 | BLEU_4 | ROUGE_L | CIDEr |
|---|---|---|---|---|---|---|
| *Concise Caption* | | | | | | |
| LLaVA1.5 (Liu et al., 2023) | 9.22 | 4.84 | 2.26 | 1.07 | 14.72 | 0.02 |
| QWEN2.5-VL (Qwen Team, 2025) | 18.42 | 9.85 | 4.90 | 2.31 | 18.12 | 2.95 |
| Geochat(Kuckreja et al., 2024) | 10.21 | 6.31 | 2.78 | 1.64 | 16.52 | 0.25 |
| GeoAssistant (Ours) | **41.64** | **29.35** | **20.16** | **14.22** | **44.04** | **25.64** |
| *Complex Caption* | | | | | | |
| LLaVA1.5 (Liu et al., 2023) | 7.13 | 3.08 | 1.11 | 0.44 | 12.06 | 0.18 |
| QWEN2.5-VL (Qwen Team, 2025) | 15.12 | 7.12 | 3.15 | 1.40 | 16.08 | 2.21 |
| Geochat(Kuckreja et al., 2024) | 14.04 | 5.64 | 2.41 | 1.18 | 14.02 | 0.22 |
| GeoAssistant (Ours) | **29.19** | **18.78** | **11.19** | **7.18** | **28.76** | **15.64** |

Table 9: Comparison of our GeoAssistant with VLMs on the SAR VQA task. OI: Object Identification, IC: Instance Counting, OC: Object Classification, OP: Object Positioning. Results are reported using the Accuracy.

| Model | OI | IC | OC | OP |
|---|---|---|---|---|
| LLaVA-1.5 (Liu et al., 2023) | 53.46 | 45.20 | 29.00 | 12.77 |
| QWEN2.5-VL (Qwen Team, 2025) | 55.46 | 47.20 | 25.00 | 13.23 |
| GeoChat(Kuckreja et al., 2024) | 62.04 | 54.36 | 51.54 | 19.84 |
| **GeoAssistant** | **77.46** | **67.24** | **62.64** | **29.72** |

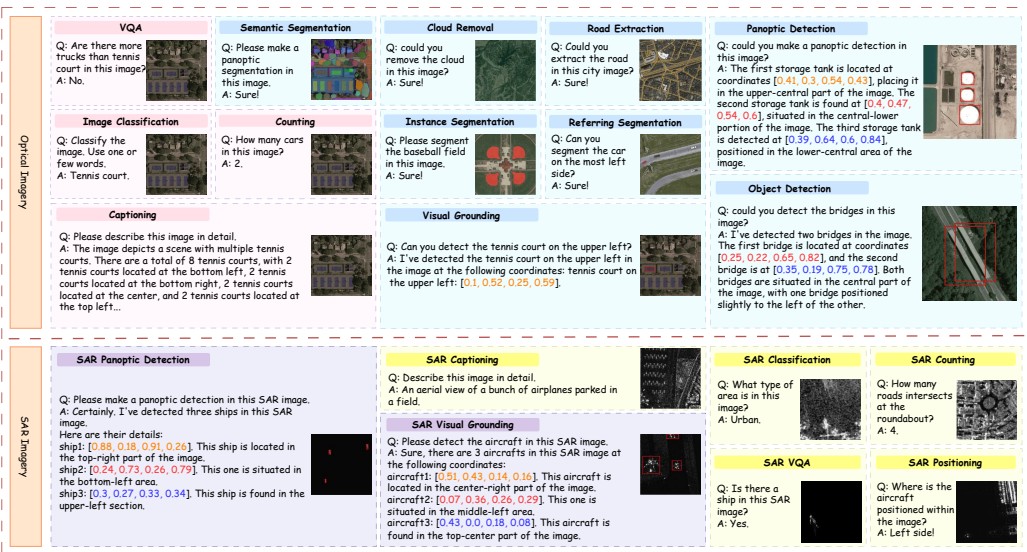

Figure 4: Qualitative results demonstrating GeoAssistant's versatility across optical and SAR imagery. The examples illustrate its ability to address both general understanding and tool-augmented tasks, highlighting robust multimodal reasoning and grounded response generation across diverse remote sensing scenarios.

## 5.4 QUALITATIVE ANALYSIS

Figure 4 presents a range of qualitative results to intuitively demonstrate GeoAssistant's capabilities. The examples in optical imagery showcase its proficiency in both holistic understanding tasks and fine-grained spatial operations. Furthermore, the model exhibits strong cross-domain performance by effectively addressing analogous tasks on challenging SAR imagery, such as SAR-specific visual grounding and panoptic detection. For instance, in the optical panoptic detection task, GeoAssistant not only identifies multiple distinct objects but also accurately provides their coordinates and spatial relationships.

## 6 CONCLUSION

In this paper, we addressed two key limitations of existing RS vision–language models: their restriction to optical imagery and lack of dynamic tool orchestration. We introduced GeoAssistant, a modular framework trained on a large-scale instruction dataset with SAR data and tool-use chains of thought. To enable systematic evaluation, we proposed GeoAssistBench, a benchmark covering optical and SAR tasks. GeoAssistant achieves strong performance across GeoAssistBench, including SAR VQA, referring object detection, and task planning. Our results show that combining a flexible tool-centric architecture with diverse instruction data and standardized benchmarks is essential for robust RS agents. This work takes a step toward practical assistants, with future directions including expanded tool repositories and improved multi-step planning.

ETHICS STATEMENT

This work adheres to the principles of open and reproducible research. GeoAssistant is built upon publicly available, open-source models, and the GeoAssistant Instruction Dataset is constructed using established open-access remote sensing datasets. Upon publication, we will release the full dataset under the Creative Commons Attribution 4.0 (CC BY 4.0) license and the source code under the MIT License. By relying solely on open resources, we aim to ensure transparency, avoid ethical concerns related to proprietary or sensitive data, and promote broad adoption for beneficial applications such as environmental monitoring and disaster response.

We acknowledge potential dual-use risks: while remote sensing technologies have valuable civilian uses, such as climate research and disaster relief, they could also be misapplied for harmful surveillance or military purposes. Our releases are intended strictly for academic research, with the goal of fostering responsible and trustworthy AI development in the remote sensing community.

REPRODUCIBILITY STATEMENT

We have taken several steps to ensure the reproducibility of our work. The architecture, training setup, and hyperparameters of GeoAssistant are described in Section 4, with additional implementation details provided in the Appendix. The construction process of our instruction dataset, including preprocessing and tool-use annotation, is documented in Section 3 and further elaborated in the supplementary materials. The datasets used in GeoAssistBench are all publicly available, and their sources and sizes are summarized in Table 3. To facilitate replication, we will release the full GeoAssistant Instruction Dataset, the GeoAssistBench benchmark suite, and the source code under open licenses upon publication. In addition, the supplementary materials already include our implementation code and a subset of the dataset. Together, these resources provide the necessary transparency for reproducing our experiments and extending our framework.

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

# A APPENDIX

## A.1 LIMITATIONS

**Manually Curated Tool Repository.** While GeoAssistant demonstrates strong performance, its current tool repertoire is manually curated. This inevitably constrains the system's adaptability to new tasks. A promising direction for future work is to develop methods for automatic discovery and seamless integration of new tools, thereby enhancing the agent's scalability and robustness.

**Modality Expansion.** Our present focus is on optical and SAR imagery, which represent two of the most widely used modalities in remote sensing. However, other data types such as hyperspectral, LiDAR, or temporal image sequences provide complementary information that is crucial in many real-world applications. Extending support to such modalities would significantly broaden the applicability of GeoAssistant and strengthen its role as a general-purpose RS assistant.

**Choice of Backbone LLM.** GeoAssistant employs Vicuna-v1.5 (7B) (Zheng et al., 2023a) as its central reasoning unit, following prior work in tool-augmented agents to ensure comparability and reproducibility. Our primary objective is not to maximize raw language performance, but to enable reliable tool selection and invocation, where Vicuna has proven effective. While this choice may somewhat limit immediate generalizability to other language or vision-language models, the framework itself is designed to be backbone-agnostic. Future extensions could flexibly incorporate stronger or more specialized models (e.g., LLaMA-3, Qwen-VL, or LLaVA-Plus), offering opportunities to further enhance reasoning capacity and multimodal integration without altering the overall architecture.

## A.2 ABLATION STUDY

To validate the effectiveness of our proposed two-stage training strategy, we conducted a series of ablation studies. The goal was to isolate the contribution of each component of our training methodology, such as the *multi-stage curriculum* and the *data mixing strategy* in the second stage. We evaluated the performance on the four sub-tasks of SAR VQA to provide a detailed analysis.

Table 10 shows the results of the ablation study on the two-stage training strategy. The results are contextualized by comparing them with the baseline models presented in our main results Table 8.

Table 10: Ablation study on the two-stage training strategy. "Multi-Stage" indicates a two-stage process, and "Stage-2 Mixing" indicates that optical data was mixed with SAR data during the second stage. Performance is reported as Accuracy (%) on the SAR VQA sub-tasks.

| Multi-Stage | Stage-2 Mixing | OI | IC | OC | OP |
|---|---|---|---|---|---|
| X | X | 55.46 | 47.20 | 25.00 | 13.23 |
| ✓ | X | 62.04 | 54.36 | 51.54 | 19.84 |
| ✓ | ✓ | **77.46** | **67.24** | **62.64** | **29.72** |

The analysis of these results clearly demonstrates the value of our approach. While a baseline single-stage training yields performance comparable to general-purpose VLMs like QWEN2.5-VL, adopting a two-stage curriculum elevates performance to a level similar to that of a strong, domain-adapted model like GeoChat. Critically, the final addition of our data mixing strategy in the second stage provides the substantial boost required to achieve state-of-the-art results. This final step is essential for mitigating catastrophic forgetting, thereby validating that the synergy between the multi-stage approach and data mixing is a key contributor to GeoAssistant's superior performance on the challenging SAR VQA task.

## A.3 METRICS APPLIED FOR EACH TASK

Here we introduced the metrics we used to measure our model's performance in each task with our motivation and their detailed formulation.

### A.3.1 ACCURACY

Accuracy is a fundamental metric used to evaluate the performance of our model across several key tasks. It generally measures the proportion of correct predictions among the total number of cases evaluated. The formula for accuracy is:

$$\text{Accuracy} = \frac{TP + TN}{TP + TN + FP + FN}$$

where:

- **TP (True Positives)**: The number of positive instances correctly identified.
- **TN (True Negatives)**: The number of negative instances correctly identified.
- **FP (False Positives)**: The number of negative instances incorrectly identified as positive.
- **FN (False Negatives)**: The number of positive instances incorrectly identified as negative.

In our work, it was applied to the following tasks: Task Planning, Visual Question Answering, and SAR Visual Question Answering, as it directly quantifies the GeoAssistant's performance on these tasks.

### A.3.2 BLEU, CIDER, ROUGE-L

To evaluate the quality of the generated text for the SAR Captioning task, we employed three standard metrics: BLEU (Papineni et al., 2002), ROUGE-L (Lin, 2004), and CIDEr (Vedantam et al., 2015). These metrics compare the machine-generated captions against human-created reference captions.

**BLEU** BLEU measures the precision of n-grams in the candidate sentence compared to the reference sentences. It calculates how many words and phrases from the generated text appear in the ground-truth text. A higher score indicates a better match. The formula is:

$$\text{BLEU} = \text{BP} \cdot \exp\left(\sum_{n=1}^{N} w_n \log p_n\right)$$

where:

- $p_n$ is the modified n-gram precision for n-grams of a specific order (the paper reports for n=1 to 4).
- $w_n$ are positive weights, typically $\frac{1}{N}$.
- BP is the Brevity Penalty, which penalizes generated captions that are too short compared to the reference length. It is calculated as:

$$\text{BP} = \begin{cases} 1 & \text{if } c > r \\ e^{(1-r/c)} & \text{if } c \leq r \end{cases}$$

(where $c$ is the length of the candidate sentence and $r$ is the effective reference corpus length).

**ROUGE-L**  ROUGE-L focuses on recall, measuring the quality of a summary by comparing it to other ideal summaries created by humans. It is based on the length of the Longest Common Subsequence (LCS) between the candidate and reference sentences. The score is calculated using an F-measure, which balances precision ($P_{lcs}$) and recall ($R_{lcs}$):

$$R_{lcs} = \frac{\text{LCS}(X, Y)}{m}$$

$$P_{lcs} = \frac{\text{LCS}(X, Y)}{n}$$

$$\text{ROUGE-L} = \frac{(1 + \beta^2) R_{lcs} P_{lcs}}{R_{lcs} + \beta^2 P_{lcs}}$$

where:

- $X$ is the reference sentence of length $m$.
- $Y$ is the candidate sentence of length $n$.
- $\text{LCS}(X, Y)$ is the length of the longest common subsequence of X and Y.
- $\beta$ is a parameter that balances the importance of precision and recall.

**CIDEr**  CIDEr is a metric designed specifically for image captioning that measures the consensus between a candidate sentence and a set of reference sentences. It performs a TF-IDF (Term Frequency-Inverse Document Frequency) weighting for each n-gram, giving more weight to informative words and less to common ones. The final score is a weighted average of cosine similarities for n-grams of different lengths. The formula for a given n-gram order $n$ is:

$$\text{CIDEr}_n(c_i, S_i) = \frac{1}{M} \sum_{j=1}^{M} \frac{g^n(c_i) \cdot g^n(s_{ij})}{||g^n(c_i)|| \cdot ||g^n(s_{ij})||}$$

where:

- $c_i$ is the candidate caption for image $i$.
- $S_i = \{s_{i1}, s_{i2}, ..., s_{iM}\}$ is the set of M reference captions for image $i$.
- $g^n$ is a vector representation of the n-grams, with TF-IDF weights.

The paper highlights that GeoAssistant achieved CIDEr scores an order of magnitude higher than the next best model, demonstrating a strong consensus with human-generated descriptions.

### A.3.3  AVERAGE PRECISION

To evaluate performance on the Referring Object Detection task, the paper utilizes the Average Precision (Acc@0.5) metric.

The core of this metric is the Intersection over Union (IoU), which calculates the ratio of the intersection area to the union area of the predicted bounding box ($B_p$) and the ground-truth bounding box ($B_{gt}$).

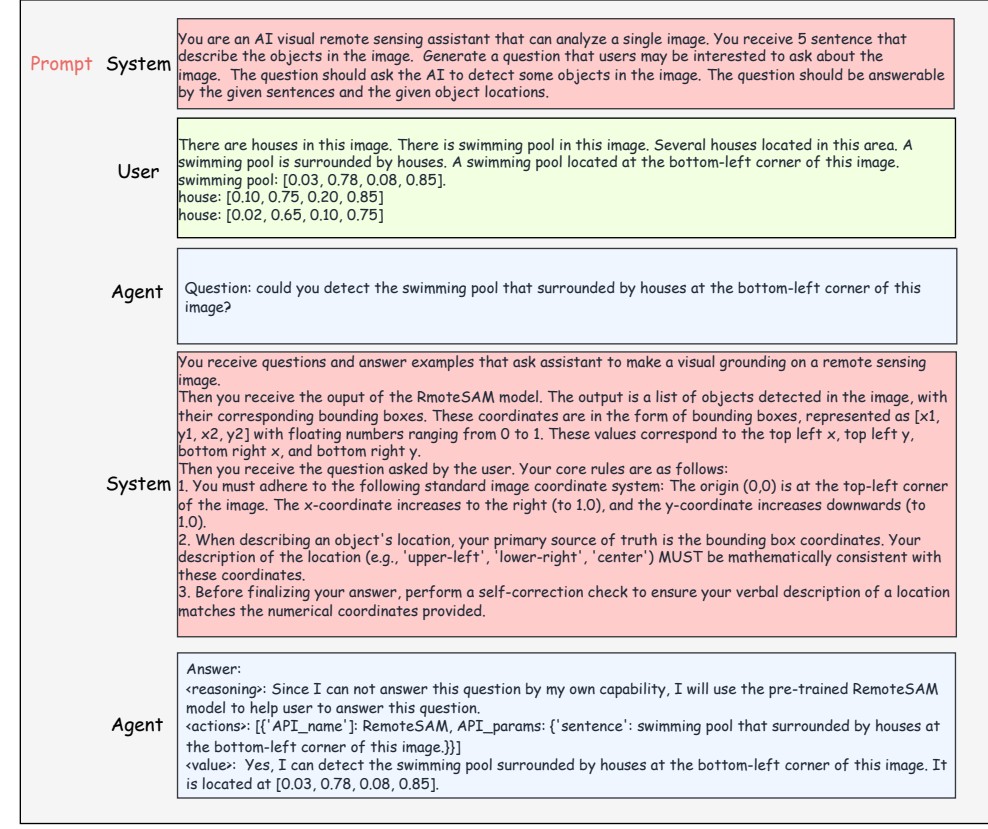

Figure 5: An example of instruction data construction using in-context learning.

The formula for IoU is:

$$\text{IoU} = \frac{\text{Area}(B_p \cap B_{gt})}{\text{Area}(B_p \cup B_{gt})}$$

A prediction is considered correct if the IoU score is 0.5 or greater. The Acc@0.5 score is then simply the percentage of predictions that meet this threshold. This provides a clear measure of how accurately the model can locate specific objects described by the user.

### A.3.4 REPRODUCIBILITY DETAILS

**Hyperparameter Settings**  Our training hyperparameters, such as the learning rate and AdamW optimizer settings, were chosen based on established practices and widely-used values for fine-tuning 7B-scale Vision-Language Models like LLaVA-v1.5.

**Decoding Strategy and Determinism**  For all evaluation tasks, we employ a deterministic greedy decoding strategy by setting the *temperature* parameter to 0. We do not use stochastic methods like beam search for our reported results. This ensures that for any given input, the model's output is identical across multiple runs. Consequently, all reported results are from a single evaluation run. Given the deterministic nature of our evaluation process, there is no statistical variation to report between runs. For the training process, we set all random seeds for relevant libraries to a fixed value of 42 to ensure the reproducibility of data processing and model initialization.

## A.4 GeoAssistant Instruction Dataset

We developed a unified data generation pipeline based on the self-instruct methodology. A key characteristic of our dataset is its unique multi-turn, tool-augmented dialogue format. Unlike traditional instruction datasets, each sample explicitly models the agent's decision-making process, including its reasoning for invoking a specific tool and the subsequent synthesis of the tool's output into a final answer. This structure is crucial for teaching the model not just what to answer, but how to leverage external capabilities. The entire dataset was generated using a carefully designed in-context learning pipeline, where a large language model was prompted to create high-quality, diverse, and plausible human-agent interactions from ground-truth labels. The role of the system prompts in this process is critical, as illustrated in Figure 5. In the first turn, a system prompt guides the LLM to formulate a realistic user query from descriptive ground-truth data. In the second, more crucial turn, the system prompt provides detailed context about the specific tool to be used. For instance, it defines the task, specifies the expected output format from the *RemoteSAM* model, establishes rules for the coordinate system, and instructs the agent to perform self-correction. This structured prompting is essential for generating the high-quality, tool-aware reasoning chains that form the core of our training data.

## A.5 Qualitative Results

To provide a more intuitive understanding of GeoAssistant's capabilities, Figure 6 presents a series of qualitative comparisons on SAR VQA tasks. The examples highlight that GeoAssistant consistently provides more accurate and contextually relevant answers compared to other leading VLMs. For instance, in object identification and counting tasks, our model correctly interprets complex SAR image features where other models often fail. This superior performance underscores the effectiveness of our training strategy and the value of our tool-augmented instruction dataset. In the spirit of transparent analysis, we also present several failure cases in the lower panel of Figure 6. These cases typically occur under conditions of high ambiguity or when dealing with object categories that are underrepresented in the training data. For example, the model may misclassify visually similar objects (e.g., a highway interchange confused with a bridge) or struggle with precise counting when objects are densely clustered. These limitations highlight a valuable direction for future work: enhancing the model's fine-grained discriminative abilities through more diverse data augmentation and potentially incorporating more sophisticated visual reasoning modules.

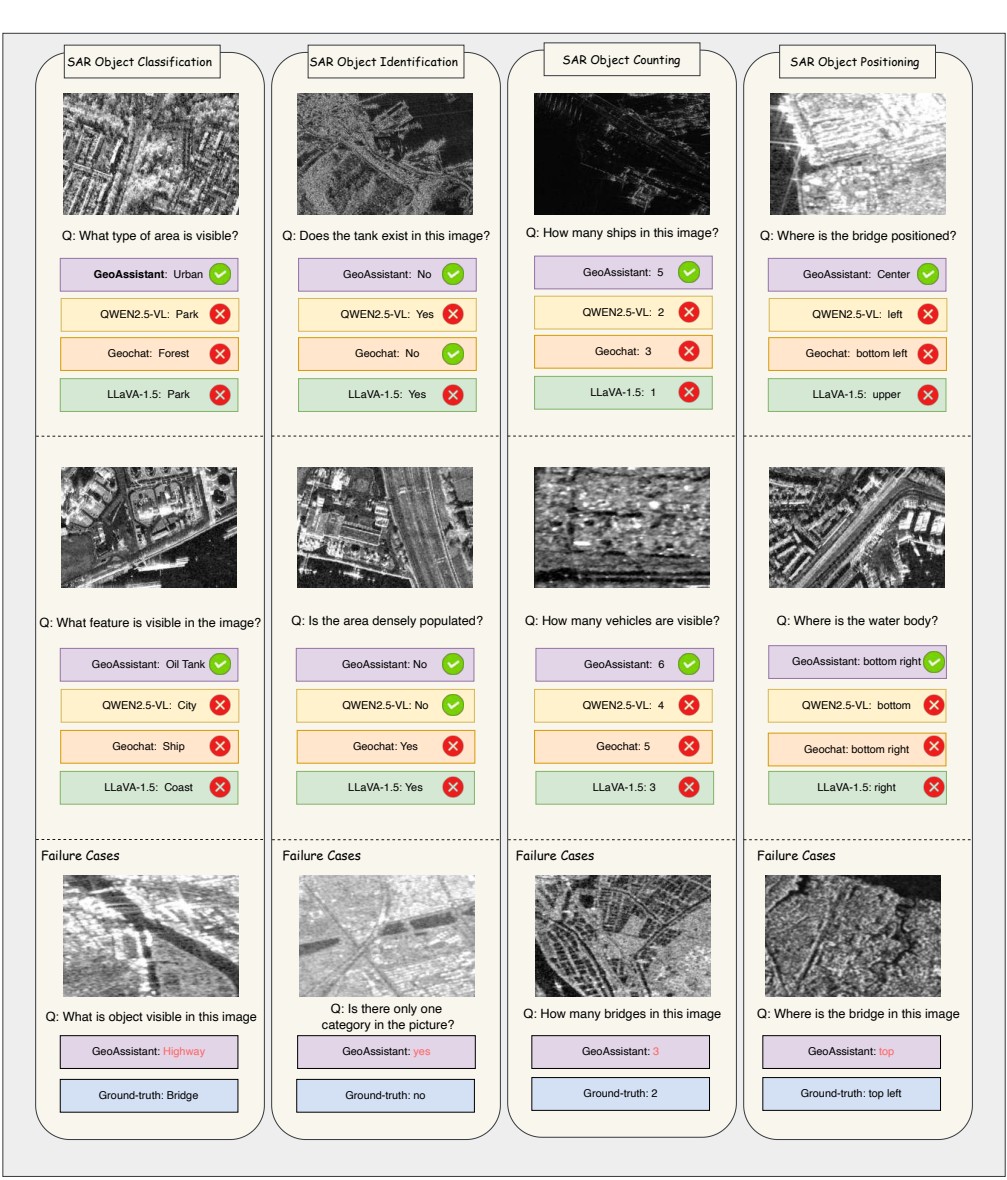

Figure 6: Qualitative results in SAR VQA tasks comparing other VLMs.

