# OpenReview forum: "GeoAssistant: A Geospatial Vision and Language Assistant that Plugs and Learns to Use Tools for Remote Sensing"
_ICLR.cc/2026/Conference — ICLR 2026 Conference Withdrawn Submission_

### Official Review · Reviewer_JqBi · 2025-10-27

**Soundness:** 2
**Presentation:** 3
**Contribution:** 3
**Rating:** 4
**Confidence:** 4

**Summary:**

This paper considers both optical and SAR imagery in remote sensing and innovatively constructs a large-scale, tool-augmented instruction dataset. Moreover, it introduces GeoAssistBench for evaluation. However, there are some weaknesses, primarily a potentially unfair experimental setup in the SAR evaluation and incomplete comparisons to other key SOTA models.

**Strengths:**

- The paper provides a comprehensive consideration of both optical and SAR imagery, addressing a critical need for all-weather remote sensing capabilities.

- It innovatively explores the construction of a large-scale, tool-augmentation-centric instruction dataset for remote sensing, integrating a diverse set of RS-specific tools.

- The proposed Agent framework successfully integrates multi-level tasks (Image-Region-Pixel) and possesses a tool-use capability that is effectively learned via end-to-end training.

**Weaknesses:**

1. Regarding Table 2, the composition of the "General Understanding" data category in Table 2 is unclear. What specific tasks or data distributions (e.g., VQA, captioning) comprise the 198K optical and 249K SAR samples?

2. Regarding Tables 5 & 6, the performance comparisons in Tables 5 and 6 (RSVQA benchmarks) are incomplete. They omit other relevant and contemporary RS VLM models such as SkySenseGPT (2024) and VHM (2025). To my knowledge, these models fit the evaluation settings (e.g., fine-tuning for RSVQA-LRBEN) and should be included for a fair assessment of the state-of-the-art.

3. The most significant weakness is the potential data leakage in the SAR-related evaluations. The proposed training dataset includes SARLANG-1M (Table 2), and the benchmark also uses SARLANG-1M for evaluation (Table 3 & 8). This means the model is tested in a "with finetune" setting. However, the baseline models it is compared against (e.g., LLaVA-1.5, Qwen2.5-VL in Table 8) appear to be evaluated in a "zero-shot" setting. This is an unfair comparison. If GeoAssistant were compared to the officially reported, fine-tuned performance of models like Qwen2-VL (from the original SARLANG-1M paper), it is questionable whether it would still show superior performance, thus casting doubt on the framework's claimed effectiveness.

4. Regarding Table 8, there is an inconsistency in the reported baseline results in Table 8. Why does the performance of LLaVA-1.5 perfectly match the official SARLANG-1M-CapBenchmark, while the performance of Qwen2.5-VL differs from it? This discrepancy requires an explanation.

**Questions:**

1. Given that this is an RS-specific model, why was the general-purpose CLIP chosen as the visual encoder instead of a domain-specific, pre-trained encoder like RemoteCLIP or SkySense-O's backbone, which are optimized for text-image alignment in remote sensing?

2. In the proposed GeoAssistBench, apart from the "Task Planning" evaluation (Table 4), are the other tasks (e.g., VQA, Referring Detection) evaluated only on the final generated answer? Or does the evaluation protocol also assess the accuracy of the intermediate steps, such as whether the correct tool was invoked by the agent to arrive at the answer?

3. Regarding the task planning accuracy in Table 4, the paper claims the 100% success rate is due to its "architectural design." I am curious what specifically causes the significant performance gap between GeoAssistant and other agents (like RS-Agent). The authors should provide a more in-depth analysis or ablation study to substantiate this claim beyond just citing the architecture.

4. I am curious if the model can collaboratively use multiple tools in a multi-turn dialogue to solve a complex problem. For example, can it first apply a de-clouding tool to a cloudy image, then execute a detection or segmentation tool on the processed image, and finally summarize the content?

5.  I think the model should learn when to call a tool and when not to. This is learned from the mix of "General Understanding" data (no tool) and "Tool-Augmented" data. The paper provides an ablation on the training strategy (Table 10) but not on the data composition. What is the impact of the data ratio (e.g., ~448K general vs. ~107K tool samples) on the agent's tendency to call (or not call) tools? An ablation showing performance with different data mixtures would be highly insightful.

---

### Official Review · Reviewer_d3M5 · 2025-10-29

**Soundness:** 3
**Presentation:** 3
**Contribution:** 2
**Rating:** 6
**Confidence:** 4

**Summary:**

This work is a multimodal vision-language assistant built for remote sensing, designed to handle both optical and SAR imagery. It aims to fill two persistent gaps in current RS models: the inability of optical-only systems to operate in diverse conditions and the lack of agents that can dynamically coordinate specialized analytical tools. Instead of relying on a fixed pipeline, GeoAssistant integrates perception, reasoning, and tool orchestration in a unified workflow that can plan, execute, and synthesize results from detectors, segmenters, or other RS tools. The model follows a structured two-turn interaction, first planning which tools to call and then generating a grounded final answer after receiving tool outputs. It is trained on a 554K sample instruction dataset combining optical and SAR data with reasoning and tool-use traces, and evaluated through GeoAssistBench, a benchmark covering multi-task and cross-domain scenarios. Experiments show gains across different tasks and planning, demonstrating a step toward AI assistants for earth observation.

**Strengths:**

- RS system to jointly handle optical and SAR imagery with dynamic tool orchestration and reasoning, overcoming the optical-only and pipeline limits of previous RS-VLMs.
- It builds a 554K-sample instruction dataset including both modalities, with explicit reasoning and tool-use traces to train multimodal, tool-aware agents.
- Introduces GeoAssistBench, a benchmark covering task planning, VQA, detection, segmentation, and SAR captioning, enabling multi-task evaluation.

**Weaknesses:**

**Limited architectural novelty.** The core framework is an adaptation of Vicuna-7B + CLIP-ViT-L/14 + MLP adaptor, following a two-turn plan-tool-answer workflow. The contribution lies in the dataset only, rather than novelty in the modeling mechanism specific to remote sensing.

**Planner evaluation is narrow.** The reported 100% planning accuracy is based on a relatively small set of about 50 query–answer pairs. This limited evaluation may inflate results and does not test generalization to paraphrased, ambiguous, or long-tail prompts.

**Data composition clarity.** The overall optical-to-SAR ratio within the ~554K instruction dataset is not explicitly reported, leaving unclear how balanced the multimodal setting actually is.

**Missing system-level metrics.** Unlike works such as GTA [1], which provide detailed tool-use and execution metrics (e.g., per-step correctness, final success), GeoAssistant omits latency, tool-argument accuracy, and failure-rate analyses, which are important for assessing deployability and efficiency of remote-sensing agents.

**Dataset quality control.** The paper does not mention any human verification, manual QA, or error-rate estimation for its synthetic instruction-tool traces. The ~554 K samples appear automatically generated, raising concerns about potential hallucinated reasoning or tool mappings that remain unquantified.

[1] Wang, Jize, et al. "GTA: a benchmark for general tool agents." Advances in Neural Information Processing Systems 37 (2024): 75749-75790.

**Questions:**

The authors are encouraged to respond to the points above to strengthen the clarity and credibility of the work. In particular, detailed clarification of the planner evaluation setup, dataset composition, verification process, and performance metrics would help better understand the practical robustness and generalizability of GeoAssistant. Please see the weakness for details.

---

### Official Review · Reviewer_mEcE · 2025-10-29

**Soundness:** 3
**Presentation:** 3
**Contribution:** 2
**Rating:** 4
**Confidence:** 4

**Summary:**

This paper proposes GeoAssistant, a tool-augmented multimodal assistant tailored for RS scenarios. It interprets user instructions, autonomously determines whether to invoke external tools, and synthesizes their outputs to generate precise responses. The innovation is that the model supports both optical and SAR images. This paper also introduces GeoAssistBench, a benchmark for cross-domain, multi-task dialogue in RS.

**Strengths:**

1. The topic of multimodal assistant tailored for RS scenarios is attractive for most readers. The overall writing is good.
2. The authors construct a large-scale instruction dataset for tool augmentation in RS, covering both optical and SAR data.
3. The experiments show some improvements over existing RS assistants across multiple tasks.

**Weaknesses:**

1. The overall contribution seems weak. The key innovation (capability to process both optical and SAR imagery) claimed the authors is too simple. Even if the existing models do not report on SAR, they can be easily extended by adding some SAR training corpus.

2. The evaluation protocol is too vague. Taking Task Planning Accuracy as an example: The benchmark only use 50 QA pairs to measure whether the agent correctly invokes the required tool (why only 50?). And with 50 QA, the accuracy should be something like "90%/92%/94%", how come the accuracy reported is "94.79%"? There are many tools involved, but only OD and IS are evaluated, why? Meanwhile, the comparison does not make sense. GPT 3.5 and 4 may not even hear of these tools, how can they answer the question? If you provide GPT with a description of each tool, the accuracy may be much higher.

3. While the authors show superior performance over other models, the performance of the tools used by GeoAssistant is not reported. It is unclear if the GeoAssistant overrides or integrates any of the results of the basic tools. If not, then GeoAssistant becomes merely a "tool selector", instead of a real assistant, which is less attractive.

**Questions:**

Refer to the questions in weaknesses.

---

### Official Review · Reviewer_Tgp2 · 2025-10-31

**Soundness:** 2
**Presentation:** 2
**Contribution:** 2
**Rating:** 2
**Confidence:** 5

**Summary:**

The paper proposed:
- A Model Product: GeoAssistant that 1. learns understanding task directly 2. learns to call extenal tools for percrption tasks 3. learns to process SAR and optical images. 4. based on LLaVA-Plus (2023)
- A tool-call data generated by Gemini 2.5 Flash, which forms the comprehensive data recipe along with other publicly available data recipes.
- A tool-use validation set of 50 samples.

**Strengths:**

- Novel Topic: Tool usage in remote sensing has not been fully explored
- Synthesizing data using the powerful Gemini 2.5 Flash, rather than open-source tools
- The paper honestly acknowledges following from Illava Plus

**Weaknesses:**

- Lack of novelity: This work follows LLaVA-Plus, but lacks of designings specifically for SAR and optical remote sensing field.
- Technically outdated: The LLaVA baseline seems outdated. There have been sar and optical vision-language models, such as EarthGPT. There are also strong general-purpose tool-use vision-langugage agent, such as Qwen2.5-VL, with advanced tool call scheme.
- Weak contribution on benchmark: Only 50 samples, and can easily be completed by existing models.
- Unfire comparison: For the task planning results, (because of the 100% acc.) the validation set is obviously overfitted on the authors' training set, which the comparison methods had not been trained on.

**Questions:**

- Comparing to LLaVA-Plus, what is novel or different in `3.1 ARCHITECTURE` and `3.2  WORKFLOW AND UPDATING FUNCTION`
- Why not call tool for all tasks (including understanding)?
- Is there any result that are actually the result of tool, not your agent? If there are, they should be annotated. (For example, maybe Table 7?)
- Figure 4 shows a large number of capabilities, but only a small number of capabilities are supported by quantitative results.
- `A.3.3 AVERAGE PRECISION` is actually `Acc@0.5` not AP. And the A.3 is not required actually.

---

### Note · Authors · 2025-12-01

I have read and agree with the venue's withdrawal policy on behalf of myself and my co-authors.